# Association of Metabolic Syndrome and Other Factors with the Presence of Diabetic Nephropathy in Type 2 Diabetic Patients

**DOI:** 10.3390/ijerph20032453

**Published:** 2023-01-30

**Authors:** Peng-Lin Tseng, Tung-Ling Chung, Chao-Hsien Lee

**Affiliations:** 1Department of Nursing, Pingtung Christian Hospital, Pingtung 900026, Taiwan; 2Department of Nursing, Meiho University, Pingtung 912009, Taiwan; 3Division of Nephrology, Department of Internal Medicine, Kaohsiung Veterans General Hospital, Kaohsiung 813, Taiwan

**Keywords:** type 2 diabetes mellitus, diabetic nephropathy, metabolic syndrome, chewing betel nut

## Abstract

Introduction: Diabetic nephropathy (DN) is a severe diabetes mellitus (DM) complication that contributes to medical and financial burdens. This study aimed to investigate risk factors for DN among type 2 diabetes mellitus (T2DM) patients by stratifying the participants based on the presence of metabolic syndrome (MetS). Materials and methods: Between June 2017 and June 2022, Taiwan Hospital was chosen for this retrospective case-control study. Following the completion of a standardized interview and the donation of blood samples for this study, participants were divided into two groups according to whether they had MetS. We contrasted how the potential DN-related factors impacted these two groups. Results: A total of 1212 patients were included, and 639 patients (52.7%) had MetS. Multivariable analysis showed that the level of educational qualifications, fasting glucose, and uric acid (UA) were associated with DN. However, chewing betel nut behavior, higher systolic blood pressure (SBP), and higher glycated hemoglobin (HbA1c) were found to be risk factors of DN among the patients who had both T2DM and MetS. Notably, betel nut chewing increased the chance of DN in T2DM patients with MetS. Conclusions: This study found that the level of education, chewing betel nut behavior, HbA1c, fasting glucose, SBP, and UA were significant risk factors for the development of DN in diabetic individuals with concurrent MetS. Our research reveals that managing the aforementioned risk factors is crucial to lowering the prevalence of DN, particularly in individuals with lower levels of education.

## 1. Introduction

The prevalence and incidence of diabetes mellitus (DM) are increasing worldwide, with an estimated prevalence of 9.3% (463 million people) in 2019 [1]. DM increases the risk of many cardiovascular complications and leads to mortality and morbidity [2]. Metabolic syndrome (MetS) and DM are both major public health issues. Those diseases have been linked to increased risk of coronary artery disease, stroke, and death [1,2]. The correlations between MetS and DM were studied due to their common adverse effects on cardiovascular systems. One of these severe complications is diabetic nephropathy (DN), which contributes to medical and financial burdens on individuals and healthcare systems. An earlier study found that MetS and obesity are separate risk factors for chronic kidney disease [3]. However, since DN was discovered, there has been no reliable evidence linking metabolic syndrome to the development of DN [4]. Moreover, the course of chronic kidney disease (CKD) is affected differently by each component of MetS, which is a mix of many components [5]. The key finding of the cohort study was that early non-diabetic CKD patients with MetS had a higher risk of CKD progression, but not late CKD patients [6]. The finding might suggest that components of MetS have a particular effect on the development of DN.

Betel quid chewing has been linked to DM, MetS, obesity, hypertension, cardiovascular illnesses, and overall mortality, according to a meta-analysis that included 17 research from Asia [7]. Chewing betel nut has been linked to obesity and weight increase in four research studies, MetS in five studies, and hyperglycemia and T2DM in two studies, according to a systematic review of eight studies [8]. In terms of MetS, a study that had 1070 Pakistanis as subjects found that chewing betel nut is unhealthy and causes MetS, especially when done in conjunction with the use of tobacco additives [9]. Studies have shown that chewing betel nuts increases the risk of T2DM, MetS, and cardiovascular events.

MetS, also known as syndrome X or the insulin resistance syndrome, was first described in 1988 by Reaven [10] and is characterized by central obesity, increased fasting blood sugar, high blood pressure, hypertriglyceridemia, and low HDL cholesterol [10]. There is mounting evidence that the MetS and each of its components contribute to the development of kidney diseases [11,12,13,14]. However, the causality is still unproven because of the complexity and inter-relationship between the components of MetS and DN. There is a sizable impact of DN on T2DM patients’ daily lives. Therefore, it is crucial to investigate the risk factors of DN and to reduce the incidence of DN. This study aimed to investigate the correlation between risk factors and DN by stratifying the participants based on the existence of metabolic syndrome.

## 2. Materials and Methods

### 2.1. Study Participants and Study Design

The case-control study was performed in the DM outpatient clinic of Pingtung Christian Hospital, a tertiary care hospital in south Taiwan, from June 2017 to June 2022. Patients were enrolled if they were older than 20 years old and have T2DM. A total of 1806 patients (age > 20) were included from June 2017 to June 2022. Patients were excluded (*n* = 594) if they had (1) non-diabetic kidney disease, (2) a history of gestational DM, (3) pregnancy during the study, or (4) the presence of serious co-morbidities (malignancy, severe infection, heart failure, liver cirrhosis, etc.) Finally, a total of 1212 patients were enrolled. Based on whether they had MetS, T2DM patients were divided into two groups (Figure 1).

### 2.2. Demographic, Medical, and Laboratory Data Collection

Each patient had a standard interview during the scheduled outpatient visit and signed the informed permission form needed to complete a brief questionnaire. Participants were seated and their SBP and DBP were recorded using a digital automated blood pressure monitor after 10 min of relaxation. The brief survey inquired about family history of DM, health habits, age, gender, educational level, weight, height, and body mass index (BMI). No qualifications, elementary school, or junior high school or higher were the three categories for educational attainment. The BMI (kg/m^2^) was calculated as the weight divided by the square of the height. Smoking, drinking, chewing betel nuts, and regular exercise were among the participants’ healthy habits. Participants who engaged in smoking behavior were those who smoked cigarettes of any kind at least once per day for three years or longer. HbA1c, fasting glucose, triglycerides, total cholesterol, high-density lipoprotein cholesterol (HDL-c), low-density lipoprotein cholesterol (LDL-c), UA, and the estimated glomerular filtration rate (eGFR) were all assessed in fasting blood samples. On an analyzer (model 7180; Hitachi, Tokyo, Japan) using high-performance liquid chromatography. Employing an enzymatic test, HbA1c, fasting blood sugar, triglycerides, total cholesterol, and HDL-c were assessed. The ratio of triglycerides, total cholesterol, and HDL-c was frequently used to calculate LDL-c indirectly. Additionally, the patient’s serum creatinine level, a trustworthy measure of renal function, served as the foundation for the eGFR value. To calculate the eGFR value, the formula 186 × creatinine^(−1.154)^ × age^(−0.203)^ (×0.742 if female) was used. For this study, DN was defined as eGFR <45 mL/min/1.73 m^2^. In addition, LDL-c was often indirectly measured using a triglyceride, total cholesterol, and HDL-c formula.

### 2.3. The Definition of Type 2 Diabetes Mellitus and Metabolic Syndrome

The participants with T2DM was defined as an HbA1c level ≥6.5%, fasting glucose level ≥126 mg/dL [15]. Based on whether they had MetS, T2DM patients were divided into two groups. According to the criteria set forth by the Health Promotion Administration of Taiwan [16], which is derived from the NCEP-ATP III guideline, patients were defined as MetS if they had any three of the following criteria: (1) Male’s waist circumstance was greater than 90 cm and female’s waist circumstance was greater than 80 cm; (2) SBP was greater than 130 or DBP was greater than 85 mmHg; (3) Fasting glucose was greater than 100 mg/dL; (4) Triglycerides was greater than 150 mg/dL; and (5) HDL-c was lesser than 40 mg/dL in men and 50 mg/dL in women.

### 2.4. Statistical Analysis

The categorical variables of T2DM patients’ characteristics—such as gender, smoking, drinking, chewing betel nuts, frequent exercise, and family history of DM—were described using absolute and relative frequency. Age, BMI, SBP, DBP, HbA1c, fasting glucose, triglyceride, total cholesterol, HDL-c, LDL-c, and UA were all continuous variables. For continuous variables, mean and standard deviation values, as well as absolute and relative frequencies, were used to describe the patient features. The normality tests of continuous variables—such as age, BMI, SBP, DBP, HbA1c, fasting glucose, triglyceride, total cholesterol, HDL-c, LDL-c, and UA—were performed by Kolmogorov–Smirnov test, and all variables underlying the dataset were found to be normally distributed. Chi-squared test was used in inferential statistics to examine the relationship between DN and T2DM patients with and without MetS. Additionally, logistic regression techniques were used to examine the relationships between the onset of DN and each of the potentially related variables in a univariate analysis and to build DN models. The best multivariable models were then selected utilizing the model selection approach after taking into account the significant factors in each statistic testing. When the *p*-value was less than 0.05, the results were considered significant and were presented as odds ratios (OR) with a 95% confidence interval (CI). IBM SPSS Statistics 24 was used to conduct the statistical analysis.

## 3. Results

### 3.1. Baseline Characteristics of All Participants

In the study, 1212 diabetic patients were recruited, with 518 women and 694 men. The mean age was 58.96 years (SD = 13.05). Among the patients, 639 patients had MetS (52.7%). A total of 278 patients had DN, of which 181 patients (28.3%) had MetS and 97 patients (16.9%) did not (OR: 1.94, 95%CI: 1.47–2.56) (Table 1). The T2DM patients who had DN have a higher relative frequency of MetS, lower educational qualifications, higher betel nut chewing rate, higher SBP, higher HbA1c, higher fasting blood glucose, and lower HDL (Table 1). Moreover, the interactions between MetS and categorical factors of DN for all T2DM patients were significant statistical in the terms of Gender×MetS (OR: 1.65, 95%CI: 1.25–2.19), Educational qualifications×MetS (OR: 1.79, 95%CI: 1.35–2.38), Chewing betel nut behavior×MetS (OR: 5.07, 95%CI: 3.44–7.49) and Family history of DM×MetS (OR: 1.47, 95%CI: 1.11–1.96). Furthermore, the interactions between MetS and continuous factors of DN for all T2DM patients were significant statistical in the terms of Age×MetS (OR: 1.01, 95%CI: 1.01–1.02), SBP×MetS (OR: 1.01, 95%CI: 1.00–1.01), DBP×MetS (OR: 1.01, 95%CI: 1.01–1.02), HbA1c×MetS (OR: 1.11, 95%CI: 1.08–1.15), Fasting glucose×MetS (OR: 1.01, 95%CI: 1.00–1.01), Triglyceride×MetS (OR: 1.43, 95%CI: 1.09–1.87), Total cholesterol×MetS (OR: 1.01, 95%CI: 1.00–1.01), HDL-c×MetS (OR: 1.01, 95%CI: 1.01–1.02), LDL-c×MetS (OR: 1.01, 95%CI: 1.00–1.01) and UA×MetS (OR: 1.17, 95%CI: 1.13–1.22).

### 3.2. Comparisons of Categorical Factors of DN with Stratification by the Presence of Metabolic Syndrome

In the MetS group, there were significant correlations between DN and educational qualifications (OR for elementary school vs. no qualifications: 0.56, 95%CI: 0.35–0.89, *p* = 0.014; OR for junior high school or higher vs. no qualifications: 0.26, 95%CI: 0.15–0.45, *p* < 0.001), and between DN and chewing betel nut (OR: 3.19, 95%CI: 1.55–6.59, *p* = 0.002) (Table 2). Similarly, as for the non-MetS group, significant correlations were found between DN and educational attainment (OR for elementary school vs. no education: 0.42, 95%CI: 0.24–0.74; OR for junior high school or higher vs. no education: 0.21, 95%CI: 0.11–0.41, *p* < 0.001), as well as between DN and betel nut chewing (OR: 4.29, 95%CI: 2.84–6.50, *p* < 0.001) (Table 2). However, among the diabetic patients with or without MetS, there was no significant correlation between DN and the other various variables (gender, smoking behavior, drinking behavior, regular exercise behavior, and family history of DM) (Table 2).

### 3.3. Comparisons of Continuous Factors of DN with Stratification by the Presence of Metabolic Syndrome

In the MetS group, the patients with DN have higher SBP levels (SBP in DN/no-DN = 147.69/144.20, SD = 15.77/11.44), higher HbA1c levels (HbA1c in DN/no-DN = 9.56/8.11, SD = 2.62/2.01), higher fasting glucose levels (fasting glucose in DN/no-DN = 213.15/167.53, SD = 98.44/74.69), and higher UA levels (UA in DN/no-DN = 7.64/6.12, SD = 2.21/1.53) than those without DN. In the univariate logistic regression analysis, the factors which were associated with the presence of DN included SBP level (OR: 1.02, 95%CI: 1.01–1.04, *p* = 0.002), HbA1c level (OR: 1.31, 95%CI: 1.21–1.41, *p* < 0.001), fasting glucose (OR: 1.01, 95%CI: 1.00–1.01, *p* < 0.001), and UA levels (OR: 1.61, 95%CI: 1.44–1.81, *p* < 0.001) (Table 3).

As for the non-MetS group, the patients with DN had higher SBP levels (SBP in DN/no-DN = 41.72/139.35, SD = 10.55/10.18), lower HDL-c levels (HDL-c in DN/no-DN = 49.18/53.12, SD = 18.91/12.51), higher fasting glucose levels (fasting glucose in DN/no-DN = 180.39/157.78, SD = 97.63/70.52), and higher UA levels (UA in DN/no-DN = 7.03/5.74, SD = 2.20/1.55) than those without DN. In the univariate logistic regression analysis, the factors which were associated with the presence of DN included SBP level (OR: 1.03, 95%CI: 1.00–1.05, *p* = 0.041), HDL-c level (OR: 0.98, 95%CI: 0.96–0.99, *p* = 0.010), fasting glucose (OR: 1.01, 95%CI: 1.00–1.01, *p* = 0.011), and UA levels (OR: 1.47, 95%CI: 1.30–1.66, *p* < 0.001) (Table 3). Besides, Box–Tidwell test were used to check the assumption of linearity to the logit for continuous independent variables in logistic regression analysis. However, all interactions between the continuous predictors and the logit (log odds) were not significant in our study (*p* > 0.05), so the assumption of linearity to the logit for continuous independent variables in logistic regression analysis was suitable.

### 3.4. Multivariable Analysis for DN Risk

We set up the DN risk models according to the presence of MetS among T2DM patients and compared the risk factors that cause DN. In the model of 573 T2DM patients without MetS, the patients who were with no qualifications (OR for elementary school vs. no qualifications: 0.49, 95%CI: 0.26–0.94, *p* = 0.032; OR for junior high school or higher vs. no qualifications: 0.28, 95%CI:0.13–0.57, *p* = 0.001), had higher fasting glucose levels (OR: 1.01, 95%CI: 1.00–1.01, *p* = 0.045) or had higher UA levels (OR: 1.49, 95%CI: 1.30–1.71, *p* < 0.001) were at an increased risk of DN.

In the model of the 639 T2DM patients with MetS, the patients who were with no qualifications (OR for elementary school vs. no qualifications: 0.45, 95%CI: 0.26–0.78, *p* = 0.005; OR for junior high school or higher vs. no qualifications: 0.19, 95%CI: 0.10–0.37, *p* < 0.001), had chewing betel quid behaviors (OR: 2.26, 95%CI: 1.32–3.86, *p* = 0.003), had higher SBP levels (OR: 1.02, 95%CI: 1.00–1.04, *p* = 0.024), had higher HbA1c levels (OR: 1.28, 95%CI: 1.15–1.42, *p* < 0.001), had higher fasting glucose levels (OR: 1.01, 95%CI: 1.00–1.01, *p* = 0.019), or had higher UA levels (OR: 1.69, 95%CI: 1.48–1.93, *p* < 0.001) were at an increased risk of DN (Table 4). A comparison of these two models revealed that the levels of educational qualifications, fasting glucose and UA were the co-risk factors for DN among T2DM patients. However, the patient’s chewing betel quid behavior, SBP, and HbA1c levels were also risk factors for DN among the T2DM patients with MetS. Betel quid chewing, especially, could exacerbate the disease condition of T2DM and even elevate the risk of DN (Table 4).

In the model of all the 1212 patients, the patient’s levels of educational qualifications, chewing betel quid behavior, SBP, HbA1c, fasting glucose, and UA were the major risk factors of DN, which was similar to the model of the 639 T2DM patients with MetS. Among them, the patients who were with no qualifications (OR for elementary school vs. no qualifications: 0.46, 95%CI: 0.30–0.71, *p* < 0.001; OR for junior high school or higher vs. no qualifications: 0.24, 95%CI: 0.15–0.38, *p* < 0.001), had chewing betel quid behaviors (OR: 2.04, 95%CI: 1.29–3.23, *p*= 0.002), had higher SBP levels (OR: 1.02, 95%CI: 1.01–1.03, *p* = 0.006), had higher HbA1c levels (OR: 1.19, 95%CI: 1.09–1.30, *p* < 0.001), had higher fasting glucose levels (OR: 1.01, 95%CI: 1.00–1.01, *p* = 0.002) or had higher UA levels (OR: 1.60, 95%CI: 1.46–1.76, *p* < 0.001) were at an increased risk of DN. However, there was no significant association between MetS and the risk of DN among T2DM patients (OR: 1.07, 95%CI: 0.76–1.49, *p* = 0.713) (Table 4).

## 4. Discussion

In this retrospective case-control study, we found that the T2DM patients with MetS have higher risk for the presence of DN. Next, we stratified analyses by the existence of MetS, and we demonstrated that lower educational status, chewing betel nut, higher SBP, increased fasting glucose, elevated HbA1c, and increased serum UA were associated with DN in diabetic patients with metabolic syndrome after controlling potentially related factors. On the other hand, in diabetic patients without metabolic syndrome, educational status, increased fasting glucose, and increased serum UA were correlated with DN after adjusting confounding factors.

Previous studies had shown the association between MetS and the onset of CKD [11,12,13,14]. In addition, many components of MetS have been reported to cause the development and progression of CKD [12,17], such as elevated BP and elevated fasting blood sugar, which is consistent with our findings. Moreover, the progress of MetS severity over time is linked to a decline in eGFR [18]. In our study, we noticed that MetS and some of its components increase the risk for DN in all diabetic patients. However, when stratifying by the presence of MetS, some factors in patients with diabetes but without MetS were not significantly correlated with DN. The potential mechanism may be explained by: (1) MetS is associated with low chronic inflammation [19], the extent of which appears to be proportional to the number of MetS components present [20]. The overt inflammation status harms the kidney. (2) Individuals with MetS may have more risk factors and an unhealthy lifestyle or behaviors, resulting in a deterioration of renal function.

Except for the components of metabolic syndrome, we also found other variables were associated with DN, including lower educational status, betel nut chewing, and increased serum uric acid. In this study, regardless of MetS status, educational level was independently linked with DN in diabetic patients. Similarly, previous studies had been reported the relationship between education level and the risk of CKD. A recent cohort study with 5095 German participants revealed that low educational attainment was positively associated with DN (OR 1.65, 95%CI: 1.36–2.0) and other adverse outcomes [21]. Another community-based longitudinal study observed lower education was associated with higher rates of incident CKD [low versus high education; hazard ratio (HR) (95%CI) 1.25 (1.05–1.48)] [22]. Further research to investigate variables that mediate the effect of educational level on DN may be necessary. In addition, our study also found elevated serum UA was positively correlated to DN. Prior studies have investigated the association between serum UA and CKD with inconsistent findings [23,24,25,26]. A prospective cohort study involving 21,475 healthy individuals reported that a modestly elevated UA level (7.0 to 8.9 mg/dL) was associated with an OR of 1.74 (95%CI 1.45 to 2.09) for new-onset CKD, while a significantly elevated serum UA level (9.0 mg/dL) was associated with an OR of 3.12 (95%CI 2.29 to 4.25) [23]. However, another cohort study with 5808 participants revealed there was no significant association between UA level and the development of CKD (adjusted odds ratio, 1.00; 95%CI, 0.89 to 1.14) [26]. The interaction of UA and the development of CKD may be explained as follows: (1) UA may be a nephrotoxin that causes kidney injury; (2) increased UA levels may aggravate other risk factors for renal function decline, particularly hypertension; and (3) UA may be a marker of other risk factors, such as metabolic syndrome and diabetes [25].

Another interesting finding of this study is that chewing betel nut is positively correlated to DN in diabetic patients with MetS after adjusting for confounding factors. However, in diabetic patients without MetS, the association was no more significant in the multivariate analysis. Chewing betel nuts, also known as betel quid or areca nuts, is widespread throughout South and Southeast Asia, East Africa, and the Western Pacific and plays an important role in the sociocultural identity [27]. Moreover, the consumption of betel quid is associated with MetS, diabetes, hypertension, obesity, cardiovascular disease [28,29,30], and cancer of the oral cavity, pharynx, and esophagus [31]. A cross-sectional research of 667 male Taiwanese participants found a link between betel nut consumption and CKD, with an adjusted odds ratio of 2.572 (95%CI 1.917, 3.451) [32], which is consistent with our finding. In our study, the patients with diabetes and MetS, the association between betel nut chewing and DN may be explained by the fact that betel nut chewing has been linked to a variety of DN risk factors. The relationship disappears in diabetic individuals who do not have MetS, and it implied that patients without MetS have a healthier lifestyle and fewer risk factors for DN development.

This study’s analysis method used stratification based on MetS status to reduce the effects from a moderating variable. However, there are some limitations to our study. First, the relatively small sample size may reduce the statistical power. Second, the study participants were recruited from a single hospital, which would not represent the general population. Third, the case-control study cannot identify the causal relationship and is more prone to bias. Fourth, the possible potential mediator between the variables and DN, such as educational level, was not examined and needs more investigation. Further research by other larger populations and longitudinal prospective studies may be required to confirm our findings.

In conclusion, this study demonstrated the risk factors associated with DN in patients with MetS or without MetS. Recognizing the risk factors is critical for optimizing DN care and improving prognosis.

### Limitations

Our study has some restrictions. Since this study was carried out in a tertiary care hospital, the participants’ enrollment may not accurately reflect all diabetes patients. By using a retrospective case-control study design, we were concerned about missing data, particularly lifestyle changes and adherence. Due to unmeasured factors, the remaining confounding factors cannot be eliminated. However, it was highly surprising to see that betel quid chewing activity was one of the possible risk factors for DN in T2DM patients, particularly in individuals with MetS, when compared to other research. To comprehend the link between DN and relative risk variables better, a larger population study is required.

## 5. Conclusions

This study demonstrated the importance of the patient’s level of education, betel nut chewing behavior, HbA1c, fasting glucose, SBP, and UA in the development of DN in T2DM patients, particularly in those with MetS. The majority of the study’s findings were in line with risk factors already reported in the literature. In addition, we discovered that the Taiwanese population’s particular risk factor is the practice of chewing betel nuts. To prevent DN, we advised patients to monitor their HbA1c, fasting glucose, SBP, UA, and betel nut chewing habits. 

## Figures and Tables

**Figure 1 ijerph-20-02453-f001:**
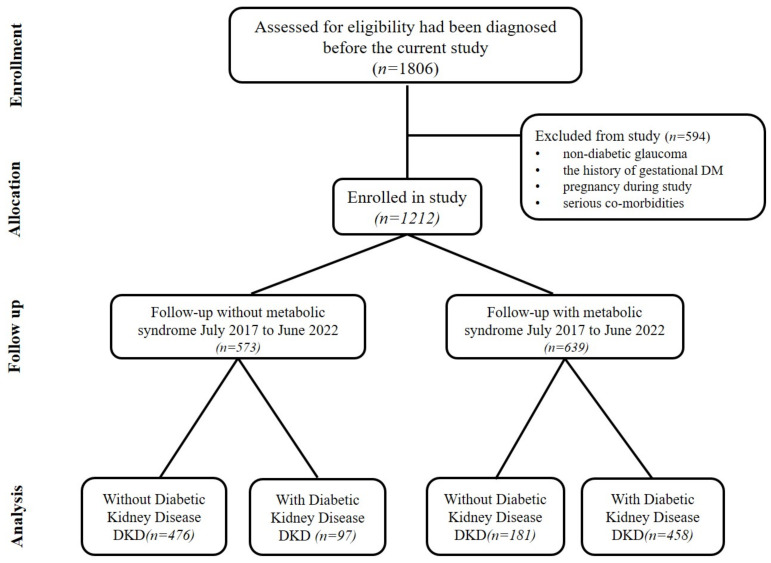
Flow diagram for the case-control study of T2DM patients in Taiwan.

**Table 1 ijerph-20-02453-t001:** Comparisons of categorical and continuous factors of DN for all T2DM patients (N = 1212).

Items		Categorical Factors (N = 1212)					Items	Continuous Factors (N = 1212)				
		Without DN (N = 934)	With DN (N = 278)	OR	95%CI			Without DN (N = 934)	With DN (N = 278)	OR	95%CI	
		Number (%)	Number (%)		Lower	Upper		Mean (SD)	Mean (SD)		Lower	Upper
MetS	No	476 (83.1)	97 (16.9)	1.94 ***	1.47	2.56	Age	58.68 (13.21)	59.89 (12.49)	1.01	0.99	1.02
	Yes	458 (71.7)	181 (28.3)				BMI (kg/m^2^)	27.71 (7.60)	27.63 (6.22)	1.00	0.97	1.03
Gender	Male	539 (77.7)	155 (22.3)	0.92	0.71	1.21	SBP (mmHg)	141.73 (10.93)	145.61 (14.44)	1.03 ***	1.01	1.04
	Female	395 (76.3)	123 (23.7)				DBP (mmHg)	81.01 (12.01)	79.41 (12.18)	0.99	0.97	1.00
Educational qualifications	No qualifications	99 (60.0)	66 (40.0)	1.00	1.00	1.00	HbA1c (%)	7.86 (1.88)	8.96 (2.51)	1.25 ***	1.18	1.33
	Elementary school	461 (75.1)	153 (24.9)	0.50 ***	0.35	0.71	Fasting glucose (mg/dL)	162.57 (72.72)	201.99 (99.21)	1.01 ***	1.00	1.01
	Junior high school or higher	374 (86.4)	59 (13.6)	0.24 ***	0.16	0.36	Triglyceride (mg/dL)	166.29 (94.38)	178.86 (94.92)	1.00	1.00	1.00
Smoking behavior	No	740 (76.9)	222 (23.1)	0.96	0.69	1.34	Total cholesterol (mg/dL)	182.07 (56.94)	184.64 (67.61)	1.00	1.00	1.00
	Yes	194 (77.6)	56 (22.4)				HDL-c (mg/dL)	47.31 (13.20)	43.02 (15.62)	0.98 ***	0.97	0.99
Drinking behavior	No	775 (77.0)	232 (23.0)	0.97	0.68	1.38	LDL-c (mg/dL)	105.74 (37.76)	106.28 (38.42)	1.00	1.00	1.01
	Yes	159 (77.6)	46 (22.4)				UA (mg/dL)	5.82 (1.55)	7.43 (2.22)	1.57 ***	1.44	1.71
Chewing betel nut behavior	No	858 (81.2)	199 (18.8)	4.48 ***	3.16	6.36						
	Yes	76 (49.0)	79 (51.0)									
Regularly exercise	No	603 (76.2)	188 (23.8)	0.87	0.66	1.16						
	Yes	331 (78.6)	90 (21.4)									
Family history of DM	No	395 (76.1)	124 (23.9)	0.91	0.70	1.19						
	Yes	539 (77.8)	154 (22.2)									

Note: *** *p* < 0.001.

**Table 2 ijerph-20-02453-t002:** Comparisons of categorical factors of DN between the patients with and without MetS (N = 1212).

Items		Without MetS (N = 573)					With MetS (N = 639)				
		Without DN (N = 476)	With DN (N = 97)	OR	95%CI		Without DN (N = 458)	With DN (N = 181)	OR	95%CI	
		Number (%)	Number (%)		Lower	Upper	Number (%)	Number (%)		Lower	Upper
Gender	Male	288 (85.2)	50 (14.8)	0.69	0.45	1.08	251 (70.5)	105 (29.5)	1.14	0.81	1.61
	Female	188 (80.0)	47 (20.0)				207 (73.1)	76 (26.9)			
Educational qualifications	No qualifications	48 (65.8)	25 (34.2)	1.00	1.00	1.00	51 (55.4)	41 (44.6)	1.00	1.00	1.00
	Elementary school	230 (82.1)	50 (17.9)	0.42 ^**^	0.24	0.74	231 (69.2)	103 (30.8)	0.56 ^*^	0.35	0.89
	Junior high school or higher	198 (90.0)	22 (10.0)	0.21 ^***^	0.11	0.41	176 (82.6)	37 (17.4)	0.26 ^***^	0.15	0.45
Smoking behavior	No	409 (83.3)	82 (16.7)	1.12	0.61	2.06	331 (70.3)	140 (29.7)	0.76	0.51	1.14
	Yes	67 (81.7)	15 (18.3)				127 (75.6)	41 (24.4)			
Drinking behavior	No	418 (82.6)	88 (17.4)	0.74	0.35	1.54	357 (71.3)	144 (28.7)	0.91	0.60	1.39
	Yes	58 (86.6)	9 (13.4)				101 (73.2)	37 (26.8)			
Chewing betel nut behavior	No	454 (84.4)	84 (15.6)	3.19 ^**^	1.55	6.59	404 (77.8)	115 (22.2)	4.29 ^***^	2.84	6.50
	Yes	22 (62.9)	13 (37.1)				54 (45.0)	66 (55.0)			
Regularly exercise	No	297 (84.6)	54 (15.4)	1.32	0.85	2.05	306 (69.5)	134 (30.5)	0.71	0.48	1.04
	Yes	179 (80.6)	43 (19.4)				152 (76.4)	47 (23.6)			
Family history of DM	No	192 (82.1)	42 (17.9)	0.89	0.57	1.38	203 (71.2)	82 (28.8)	0.96	0.68	1.36
	Yes	284 (83.8)	55 (16.2)				255 (72.0)	99 (28.0)			

Note: ^*^
*p* < 0.05, ^**^
*p* < 0.01, ^***^
*p* < 0.001.

**Table 3 ijerph-20-02453-t003:** Comparisons of continuous factors of DN between patients with and without MetS (N = 1212).

Items	without MetS (N = 573)					with MetS (N = 639)				
	Without DN (N = 476)	With DN (N = 97)	OR	95%CI		Without DN (N = 458)	With DN (N = 181)	OR	95%CI	
	Mean (SD)	Mean (SD)		Lower	Upper	Mean (SD)	Mean (SD)		Lower	Upper
Age	59.43 (13.71)	61.25 (12.58)	1.01	0.99	1.03	57.91 (12.64)	59.16 (12.41)	1.01	0.99	1.02
BMI (kg/m^2^)	25.81 (8.44)	26.85 (8.04)	1.01	0.98	1.05	29.22 (6.48)	27.97 (5.29)	0.96	0.91	1.01
SBP (mmHg)	139.35 (10.18)	141.72 (10.55)	1.03 *	1.00	1.05	144.20 (11.14)	147.69 (15.77)	1.02 **	1.01	1.04
DBP (mmHg)	73.96 (9.10)	73.80 (7.85)	1.00	0.96	1.04	84.71 (11.71)	85.01 (14.74)	1.00	0.98	1.02
HbA1c (%)	7.62 (1.71)	7.84 (1.82)	1.07	0.95	1.20	8.11 (2.01)	9.56 (2.62)	1.31 ***	1.21	1.41
Fasting glucose (mg/dL)	157.78 (70.52)	180.39 (97.63)	1.01 *	1.00	1.01	167.53 (74.69)	213.15 (98.44)	1.01 ***	1.00	1.01
Triglyceride (mg/dL)	129.41 (75.49)	139.05 (80.98)	1.00	1.00	1.00	206.90 (96.48)	203.02 (93.88)	1.00	1.00	1.00
Total cholesterol (mg/dL)	177.31 (44.45)	179.36 (43.64)	1.00	1.00	1.01	187.02 (67.21)	187.46 (77.43)	1.00	1.00	1.00
HDL-c (mg/dL)	53.12 (12.51)	49.14 (18.91)	0.98 *	0.96	0.99	41.26 (11.01)	39.74 (12.40)	0.99	0.97	1.00
LDL-c (mg/dL)	104.93 (35.26)	106.22 (38.36)	1.00	1.00	1.01	106.58 (40.22)	106.32 (38.56)	1.00	1.00	1.00
UA (mg/dL)	5.74 (1.55)	7.03 (2.20)	1.47 ***	1.30	1.66	6.12 (1.53)	7.64 (2.21)	1.61 ***	1.44	1.81

Note: * *p* < 0.05; ** *p* < 0.01; *** *p* < 0.001; Box–Tidwell test for all interactions between the continuous predictors and the logit (log odds) were not significant.

**Table 4 ijerph-20-02453-t004:** Comparisons of risk factors of DN between type 2 DM patients with and without MetS.

Models	Without MetS (N = 573)			With MetS (N = 639)				All Data (N = 1212)		
Variables (Reference)	OR	95%CI	*p*-Value	OR	95%CI	*p*-Value	OR	95%CI	*p*-Value	VIF
MetS (No)							1.07	[0.76, 1.49]	0.713	1.34
Educational qualifications (No educational qualifications)										
Elementary school	0.49 *	[0.26, 0.94]	0.032	0.45 **	[0.26, 0.78]	0.005	0.46 ***	[0.30, 0.71]	<0.0001	2.37
Junior high school or higher	0.28 **	[0.13, 0.57]	0.001	0.19 ***	[0.10, 0.37]	<0.001	0.24 ***	[0.15, 0.38]	<0.001	2.38
Chewing betel nut behavior (No)	1.73	[0.62, 4.83]	0.297	2.26 **	[1.32, 3.86]	0.003	2.04 **	[1.29, 3.23]	0.002	1.33
SBP	1.02	[0.99, 1.05]	0.120	1.02 *	[1.00, 1.04]	0.024	1.02 **	[1.01, 1.03]	0.006	1.10
HbA1c	1.03	[0.87, 1.22]	0.735	1.28 ***	[1.15, 1.42]	<0.001	1.19 **	[1.09, 1.30]	<0.001	1.53
Fasting glucose	1.01 *	[1.00, 1.01]	0.045	1.01 *	[1.00, 1.01]	0.019	1.01 **	[1.00, 1.01]	0.002	1.26
UA	1.49 **	[1.30, 1.71]	<0.001	1.69 **	[1.48, 1.93]	<0.001	1.60 **	[1.46, 1.76]	<0.001	1.06

Note: * *p* < 0.05; ** *p* < 0.01; *** *p* < 0.001.

## Data Availability

Data are contained within the article.

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
