# Peer review of "Association of Metabolic Syndrome and Other Factors with the Presence of Diabetic Nephropathy in Type 2 Diabetic Patients"

_ijerph, 2023, doi:10.3390/ijerph20032453_

Round 1

Reviewer 1 Report

This manuscript, a retrospective case-control study performed at a single institution, aims to assess the risk factors for diabetic nephropathy in individuals with type 2 diabetes stratified by the presence or absence of the metabolic syndrome. The authors report on several risk factors in individuals with type 2 diabetes for the presence of diabetic neuropathy, including the presence of the metabolic syndrome. Additionally, the authors report on risk factors for the presence of diabetic nephropathy in those with metabolic syndrome and those without metabolic syndrome. This study does allow for the assessment of these risk factors, and a strength of the study is the stratification of those with and without the metabolic syndrome. The majority of these risk factors, however, are fairly well-reported as known risk factors for microvascular diabetes-associated complications such as diabetic nephropathy. I also struggled with several other aspects of the study, as outlined below: 

1)    The first several sentences of the introduction paragraph (lines 30-36) use past tense, which reads awkwardly as, even though the epidemiological studies these statements are based off of were performed a few years ago, we assume they still hold true. For example, “Metabolic syndrome and diabetes mellitus were both major public health issues” makes it sound as if they no longer are. 

2)    A few statements would benefit from a source citation, as outlined below:

a.     Line 39, “an earlier study found that MetS and obesity are separate risk factors for chronic kidney disease” 

b.     Line 43, “The key finding of the cohort study was that early non-diabetic CKD patients with MetS had a higher risk of CKD progression…”

c.     Line 61, “…the primary cause of DN is insufficient diabetes management”

3)    Important to define how individuals were identified as having type 2 diabetes (Line 70). Was this via clinical diagnosis based on chart review, insulin resistance testing (HOMA-IR, etc.), negative islet cell antibodies, etc. 

4)    Results 3.1 (Line 131)-The authors report results for both chi-squared test and odds ratio. While both are tests of association, it is confusing to the reader to mix the two. One of these two should be chosen to be represented in the table (table 1) and then discussed in the results, either chi-square testing (test of association) or OR (measure of association). This will make this section easier to read, relate easily to table 1 and demonstrate that the authors are consistently answering the same question of association. 

5)    Table 1 should be cleaned up regarding rows so it easier to follow. Bottom left corner: align the ‘yes’ and ‘no’ better with the item (such as drinking behavior). On the right-hand side of the table make all of the rows even (some of the rows have a ‘)’ that has been pushed to the next row which makes it challenging to follow for the reader

6)    Again, issue between using both chi-square and OR testing for the same question, in results section 3.2 in line 140 the authors state, “…and a higher rate of chewing betel nut” and in line 146 the authors report the association as “correlation”. Should stay consistent in either comparing rates or frequencies or reporting correlation. 

7)    One question I had. As the analysis was stratified by presence or absence of MetS, are there any concerns for confounding by subsequently analyzing risk factors (such as fasting glucose) which also make up the diagnosis of MetS? A comment from the biostatistician may be helpful. 

8)    The major interest of the study involves the stratification of the analysis by MetS status, however, the majority of the risk factors analyzed are well-known to impact microvascular diabetes-associated complications such as diabetic nephropathy. 

9)    Minor grammatical aspects:

a.     Line 50, “…and weight increase in four research, MetS in five studies…”, should say “…and weight increase in four research studies”

b.     Line 64, “…study aimed to investigate the correlation between the risk factor and DN by stratifying.” Should say “risk factors” instead of “the risk factor” as more than one risk factor was assessed 

Reviewer 2 Report

Peng-Lin Tseng et al., in " Association of metabolic syndrome and other factors with the presence of diabetic nephropathy in type 2 diabetic patients " provide a retrospective observational analysis, in a case-control study design. They outline that patient's level of education, betel nutchewing behavior, HbA1c, fasting glucose, SBP, and UA were significantly associated with the development of DN in T2DM patients, particularly in those with MetS.

 Despite the potential interesting topic, the paper suffers from some issues related to the methodology used in data analysis and discussion of the results.

Here are some comments to consider in order to improve the content of this manuscript:

1.  In the Abstract section , the authors stated that the aim of the study was to investigate the risk factor for Diabetic nephropathy (DN) among type 2 diabetes mellitus (T2DM) patients with or without Metabolic syndrome (MetS) and the interrelationship between MetS and DN-related risk factors but but the last part of the objectives is not found in the Results section.

 2.   In Statistical Analysis section, the authors stated that for continuous variables, mean and standard deviation values were used. Why were used mean and standard deviation as measures of centrality for distributions of continuous variables?

 3.     In Statistical Analysis section, the authors stated that logistic regression techniques were used to examine the relationships between the onset of DN and each of the potentially related variables in a univariate analysis and to build DN models. What methods were used to check the assumption of linearity to the logit for continuous independent variables in logistic regression analysis?please report the results in the legend of Table 3,

 4.     In the Results section, the authors stated that T2DM patients with DN had a higher prevalence of MetS but in a retrospective case-control study design with a sample selected from a single-center DM outpatient clinic the prevalence cannot be estimated (only relative frequency of disease).

 5.     Comparisons of potential (categorical or continous) factors of DN with stratification by the presence of metabolic syndrome assume that the authors found a signifiant statistical interaction between MetS and studied factors in all  sample of T2DM patients? Please report the results of interaction effects in the section related to the comparisons of categorical and continuous factors of DN for all T2DM patients.

 6.     In the multivariable model of DN tested in all sample (n=1212) there might be aspects related to multicollinearity between indepedent variables (MetS, Fasting glucose, SBP). Please report performance measures for all multivariable tested models ( in the legend of Table 4).

7.     In the Results section , all tables with the results of the regression analysis should have the same format for reporting the results.

8.     Discussion must be improved. Although the authors stated that MetS was associated with the presence of DN after adjusting significant factors, the results described in Table 4 showed that there was no significant association between MetS and DN after adjusting for other covariates (p=0.713).

9.     Althought the authors declared that stratification based on MetS status was used to reduce confounding effects, it seems rather that MetS is a moderating variable rather than a confounding factor.

10.  References 8 and 9 represent the same reference? Please verify...

Round 2

Reviewer 1 Report

The authors made acceptable edits to the manuscript to address the major concerns that were raised upon initial review. 

Reviewer 2 Report

I think that this modified version brought significant improvements but there are still some aspects to improve in the data analysis methodology:

1.      In the letter containing the authors' responses, the definitions of the centrality parameters were justified. We know these definitions, however i have just one question if authors can explain which test they use to determined that that continuous variables follow Normal distribution? please add response in Statistical Analysis section

2.      in the Results section, the authors inserted statements without numerical results (please see lines 178-182,  where are the p values?!)
